# Parametric Study on Microwave-Assisted Pyrolysis Combined KOH Activation of Oil Palm Male Flowers Derived Nanoporous Carbons

**DOI:** 10.3390/ma13122876

**Published:** 2020-06-26

**Authors:** Napat Kaewtrakulchai, Kajornsak Faungnawakij, Apiluck Eiad-Ua

**Affiliations:** 1King Mongkut’s Institute of Technology Ladkrabang, College of Nanotechnology, Bangkok 10520, Thailand; knapat.kara@gmail.com; 2National Nanotechnology Center (NANOTEC), National Science and Technology Development Agency (NSTDA), Pathum Thani 12120, Thailand; kajornsak@nanotec.or.th

**Keywords:** oil palm male flowers, nanoporous carbons, microwave-assisted pyrolyzer, KOH activation

## Abstract

Oil palm male flowers (PMFs), an abundant agricultural waste from oil palm plantation in Thailand, have been utilized as an alternative precursor to develop nanoporous carbons (NPCs) via microwave-assisted pyrolysis combined potassium hydroxide (KOH) activation. The influences of relevant processing variables, such as activating agent ratio, microwave power, and activation time on the specific pore characteristics, surface morphology, and surface chemistry of PMFs derived nanoporous carbons (PMFCs) have been investigated to explore the optimum preparation condition. The optimum condition under a microwave radiation power of 700 W, activation holding time of 6 min, and activating agent ratio of 2:1 obtained the PMFC with the highest Brunauer–Emmett–Teller (BET) surface area and total pore volume approximately of 991 m^2^/g and 0.49 cm^3^/g, composed of a carbon content of 74.56%. Meanwhile, PMFCs have a highly microporous structure of about 71.12%. Moreover, activating agent ratio and microwave radiation power indicated a significant influence on the surface characteristics of PMFCs. This study revealed the potential of oil palm male flowers for the NPCs’ production via microwave-assisted KOH activation with a short operating-time condition.

## 1. Introduction

Nanoporous carbons (NPCs), one of carbon materials with a highly porous structure and enlarged surface area, have been conventionally produced from natural coal or petroleum solid wastes. The utilization of NPC is extremely useful in adsorption and separation processes for contaminant removal because of its high adsorptive performance [1]. Recently, it was noted to conduce to energy applications such as biocatalyst and catalyst support material [2,3], as well as electronic devices [4]. NPC applications in catalysis pathways are served owing to a favorable pore structure, extensive degree of surface active-site, and high surface area. However, NPC produced from various types of agricultural residues that could be applied as a metal catalyst support for the biofuel and biochemical production. Additionally, recent studies explored the use of NPC as a high-performance supercapacitor electrodes [5].

The production of NPC from bio-wastes is commonly employed using two steps, that is, a carbonization followed by activation. The two categories of activation are known as physical activation and chemical activation [6]. Basically, physical activation involves a partial gasification of char at high activation temperature (600–1100 °C) using steam, carbon dioxide, or the mixture of these two gases as an activating agent [7]. Meanwhile, in chemical activation, char precursor is mixed with chemicals such as sodium hydroxide (NaOH), potassium hydroxide (KOH), zinc chloride (ZnCl_2_), and phosphoric acid (H_3_PO_4_), and then substituted to a carbonization. The pore development exhibits from dehydration and degradation of carbon precursor [8,9,10]. Technically, the physical activation usually conducts at a higher temperature and longer period of treatment time as compared with the chemical activation owing to high energy required in the reaction of char with steam or carbon dioxide. Therefore, chemical activation may reduce the energy consumption by minimizing the reaction temperature [11]. A large surface area and high production yield were found in the product obtained by chemical activation [12,13]. However, KOH has become a popular activating agent as it provides some advantages, such as the development of a narrow pore structure [14]. According to the heating process, there are three main heating processes for NPC production (i.e., heating from partial combustion of feedstock, heating from external source, and microwave heating) [15]. The most popular in an industrial scale is heating from a partial combustion thanks to simple operation, however, the production yield is significantly reduced by partial burning of precursor. Nevertheless, the heating from external source is commonly used in a laboratory or pilot scale. This technique requires high energy consumption and a longer period of operation time. In addition, the microwave heating system becomes an effective process for conversion of biomass to NPC owing to rapid heating with a lower operation time, resulting in a reduction of energy consumption. In the microwave system, heat is provided by the direct interaction between microwave and materials, thus leading to the uniform temperature for the reaction. Furthermore, the materials that are applied under the microwave heating system should be a dielectric material such as carbon because of a high performance interaction for heat generation [16].

The specific characteristics of NPC, which can be controlled by the manufacturing, are influenced by its application. The properties, quality, and cost of NPC also depend on the raw feedstocks. Specific characteristics of precursors such as rich carbon content and high-density are essential for the NPC production, leading to interesting features including high production yield, greater porosity, and good mechanical property [17,18]. However, coal-based precursor is quite expensive for commercial NPCs. Therefore, the utilization of agricultural wastes has been widely investigated to supersede the position of coal or petroleum solid wastes thanks to renewable and sustainable material, cheapness, carbon sequestration, and abundance in nature. In recent years, varied studies have readily revealed that available relevant by-products from agricultural activities such as coconut shells [19], palm kernel shells [20], palm woods [21], date stones [22], pistachio nut shells [23], coffee residues [24], and tobacco stems [25] have been used as a precursor for porous carbon production. Moreover, waste water sludge and other organic substances were successfully applied [26]. However, oil palm is one of the world’s most significant crop, which tends to grow continuously in the future owing to a high demand in food and biorefinery industries [27,28]. Each part of oil palm can be beneficially utilized in various industries and laboratories such as palm oil, palm kernel oil, oil palm shells, empty fruit brunches, and leaves [29,30]. Meanwhile, oil palm male flowers, an interested agricultural waste, which have large amounts in cultivated and plantation areas, have not been widely investigated for NPC manufacturing. Although, a recent study reported that oil palm male flowers have been developed into a hydrochar via hydrothermal process. This hydrochar has a low specific surface area and poor carbon content compared with NPC, which may not suitable for promising applications such as an efficient adsorbent and catalyst support [31]. For this reason, the motivation of this study is to develop an oil palm male flower into NPC. Here, we reported that oil palm male flowers derived nanoporous carbon (PMFC) was obtained via microwave-assisted pyrolyzer system with chemical activation using KOH as an activating agent. Accordingly, the effects of preparation parameters, such as microwave radiation power, activation time, and activating agent ratio (carbonized PMFs/KOH, w/w) on the physical and chemical characteristics of PMFCs, such as pore characteristic, chemical content, elemental composition, and surface chemistry and morphology, were obviously characterized and discussed.

## 2. Materials and Methods

### 2.1. Materials

Oil palm male flowers (PMFs), collected from a local plantation in southern of Thailand, were crushed and sieved into the approximate sizes of 0.5–3 mm. The proximate and ultimate analysis of oil palm male flowers (PMFs). The proximate analysis showed that PMFs feedstocks have a fixed carbon of 24.01% and volatile matters of 61.06% (Table 1). Potassium hydroxide (KOH) and hydrochloric acid (HCl), laboratory grade, were supplied from CARLO ERBA Reagents Co., Ltd., (Paris, France). High purity-grade (99.99%) nitrogen was used in an experiment.

### 2.2. Preparation of Oil Palm Male Flowers Derived Nanoporous Carbon (PMFC)

Firstly, the prepared PMFs were carbonized into a char at 500 °C with a heating rate of 10 °C/min for 1 h in a horizontal stainless-steel tube reactor to enhance a carbon content owing to a greater dielectric property of carbon materials as raw PMFs show absolutely poor interaction with microwave radiation [15]. The carbonized PMFs were physically mixed with KOH at varied weight ratios of 1:1, 2:1, 4:1, 6:1, and 8:1 (PMFs char/KOH) Additionally, the immerged PMFs chars were then substituted to KOH activation in a quartz tube (38 mm, O.D.) on a custom-made microwave-assisted pyrolyzer at a frequency of 2.45 GHz (Samsung MS23F301EAW, Bangkok, Thailand, 800 W full heat power). The effects of microwave radiation powers (450, 600, 700, and 800 W) were carefully investigated by constantly fixing microwave radiation time at 6 min using an activating agent ratio of 2:1, while the different microwave radiation times were studied from 4 to 12 min based on the preliminary run at 700 W, and the activating agent ratio was also kept at 2:1. After the microwave-assisted KOH activation, the resulting PMFC was cooled to room temperature in a nitrogen atmosphere. Then, the prepared samples were washed with 0.5 M HCl and deionized water several times until completely neutral pH. The washed samples were dried at 105 °C for 12 h, crushed, and sieved to 0.5 mm for a uniform particle size. PMFC was noted that C followed by microwave radiation power, time, and activating agent ratio. Herein, PMFC450-6-2:1 referred to PMFC prepared at a microwave radiation power of 450 W for 6 min using an impregnation ratio of 2:1. The PMFC yield was calculated based on dry basis by the following equation:(1)$$\mathrm{PMFC}\;\mathrm{yield}=\frac{\mathrm{Mass}\;\mathrm{of}\;\mathrm{PMFCs}}{\mathrm{Mass}\;\mathrm{of}\;\mathrm{raw}\;\mathrm{PMFs}}\;\times \;100$$

### 2.3. Characterization

#### 2.3.1. Proximate and Ultimate Analysis

Proximate analysis is the technique separating the chemical compositions into four portions including moisture, volatile substance, fixed carbon, and ash content [8,32]. The moisture was determined by a drying process following American Society for Testing Materials (ASTM) D2867-99 (ASTM, 2014) [33]. Volatile substances were measured by heating process under an inert atmosphere, ASTM D5832-98 (ASTM, 2008) [33]. Ash content was analyzed by direct combustion follow ASTM D2866-94 (ASTM, 2011) [34]. Meanwhile, fixed carbon was calculated by subtracting from the other compositions [35]. Ultimate analysis reveals the elemental compositions of raw PMFs and as-performed samples (i.e., carbon, hydrogen, and nitrogen) were determined using a CHN (Carbon, Hydrogen, Nitrogen) elemental analyzer (Leco truespec chns-628). The oxygen percentage was directly calculated by the difference of all elemental compositions from 100% [34,35].

#### 2.3.2. Surface Characteristics

Porosity and pore structure of as-performed PMFCs were carried out using nitrogen adsorption–desorption analysis measured by Micromeritics TriStar II 3020 surface area analyzer (Norcross, GA, USA) operating at −196 °C. The Brunauer–Emmett–Teller (BET) method was used for determining the BET surface area (S_BET_) [31]. Pore size distribution was analyzed using the Barrett–Joyner–Halenda (BJH) model [7]. The total pore volume (V_T_) was evaluated by a condensation of liquid nitrogen at the relative pressure (*P/P_0_*) of 0.99. Meanwhile, the micropore volume (V_mic_) was determined by the t-plot model [8,36]. Mesopore volume (V_mes_) was calculated by subtracting micropore volume based on total pore volume [8,37].

#### 2.3.3. Surface Functional, Crystallinity, and Morphology Analysis

The surface functional characteristics of raw PMFs and as-performed PMFCs were studied using Fourier transform infrared spectroscopy (FTIR). The infrared absorption spectra are in the wavenumber ranges from 4000 to 400 cm^−1^ in transmittance mode using a Perkin Elmer UATR (Universal ATR) Two (Waltham, MA, USA) [35,38].

X-ray photoelectron spectroscopy (XPS) of PMFCs was conducted using a PHI5000 Versa Probe II (ULVAC-PHI Inc., Kanagawa, Japan) with Al Ka (1.486 keV) as an excitation beam at beam line 5.3, the SUT-NANOTEC-SLRI joint research facility, Synchrotron Light Research Institute (Public Organization), Nakhon Ratchasima, Thailand. The samples were situated on the carbon tape placed on the steel stub and substituted to high vacuum system (1 × 10^−8^ mbar) for 2 h before measurement. All binding energy spectra were processed by Multipak software to fit the desired spectra (C1s and O1s contributions) [5,10].

The crystallinity for PMFCs was determined by Raman spectroscopy using Thermo scientific DXR SmartRaman (Waltham, MA, USA). The spectra were measured at constant room temperature using a wavelength of 532 nm recorded from 500 to 2500 cm^−1^ [39].

The surface morphology was observed by scanning electron microscope (SEM) on Zeiss EVO50 (Oberkochen, Germany). The samples were sprinkled on a carbon tape located on steel sample holder and coated by gold sputtering to enhance electron conductivity for identification [36].

## 3. Results and Discussion

### 3.1. The Effects of Preparation Variables on Production Yield of PMFCs

Figure 1a demonstrates the yield of carbonized PMFs at 500 °C for 1 h before further use in the activation experiment. The PMFs char yield is approximately 33.5 wt% owing to the decomposition of organic substances in biomaterial structures, which are commonly composed of cellulose, hemicellulose, and lignin. The influences of preparation variables on production yield of PMFCs are displayed in Figure 1b–d, respectively. In Figure 1b, an increase in an activating agent ratio tends to decrease the PMFC production yield because a rise in the activating agent ratio leads to a high activation degree. Furthermore, the carbon structure is more reacted and decomposed following the study of Shijie et al. [12]. They also reported that a higher activating agent ratio significantly decreased the yield of gulfweed activated carbons. Moreover, an increase in microwave radiation power obviously decreases the PMFC yield, as illustrated in Figure 1c. Foo and Hameed [40] showed the effects of microwave powers in a preparation of activated carbon from biodiesel solid residue. BET surface area might develop by enhancement of microwave power. The porous carbon yield is gradually decreased as a higher microwave radiation power serves more heat energy for the activation. Therefore, the carbon atoms are continuously consumed during the activation by thermal decomposition. Figure 1d displayed the effects of activation times on the PMFC yield. An increment of activation time achieves a reduction of PMFC yield until the activation time rises to 8 min. After 8 min, the PMFC yield was slightly decreased because of the completed activation between activating agent with carbon atoms [41].

### 3.2. Surface Morphology

The surface morphology observation of both carbonized PMFs and PMFC was carried out via using a scanning electron microscopy (SEM) technique at a magnification of 500×. Figure 2a displays the SEM micrograph of carbonized PMFs. It can be seen that there is no porous structure on the carbonized PMFs, and the external surface is quite smooth and dense. Figure 2b and c showed the external surface of PMFCs (PMFC700-4-2:1 and PMFC700-6-2:1) with the BET surface area of about 911 and 991 m^2^/g, respectively. Although, the pore cavities observed by SEM (Figure 2b,c) are a microstructure of the external pores of PMFC. The PMFC has various pores and cavities on the external PMFC surface, which are caused by microwave-assisted KOH activation. However, this pores structure can possibly be used to identify the porous carbon structures. These extensive external pores are the entry way into internal mesopores an micropores [32]. In particular, KOH reacting with the carbon atoms of PMFs char during activation leads to the formation of porous structures including micropores, mesopores, and external pores. Pore structures are significantly impacted by the different operating conditions during microwave-assisted KOH activation, such as activation time and microwave radiation power [17]. However, Figure 2d shows high pore cavities observed by the cross-section image of PMFCs. The observation on the PMFC surface characteristic is related to the textural pore characteristic of produced PMFC. The results revealed that high porosity represented various pores on PMFCs’ surface.

### 3.3. The Effects of Experimental Variables on Surface Area

This investigation shows the influences of different experimental variables on the BET surface area. It can be seen that activating agent ratio and microwave radiation power exposed the supreme influences. Moreover, the activation holding time has a significant impact on the BET surface area.

#### 3.3.1. Effect of Activating Agent Ratios on BET Surface Area

Figure 3 present the relevant factors impacted on the BET surface area of PMFCs. More specific information is straightly explained. In Figure 3a, the investigation of activating agent ratios of 1:1, 2:1, 4:1, 6:1, and 8:1 (carbonized PMFs: KOH, w/w) on the BET surface area are demonstrated by keeping the condition at a microwave radiation power of 450 W for 6 min. The results revealed that the optimum activating agent ratio is 2:1 to achieve the highest BET surface area. The PMFC exhibits the BET surface area over 750 m^2^/g. According to the activating agent ratio (PMFs char/KOH) rising from 1:1 to 2:1, the BET surface area was gradually increased as the activating agent is more at an excess at the 1:1 activating agent ratio. This is because of high activation degree occurring, while the pores were completely formed by the activation reaction between excess KOH and carbon atoms. When the KOH still reacted with carbon atoms after the pore formation, it significantly caused a broken pore structure resulting in the decrease in BET surface area [42] After that, the activating agent ratio was above 2:1, and the BET surface area was decreased rapidly and continued slightly lower because KOH molecules used in activation were not excessive (Figure 3a) [35,43].

#### 3.3.2. Effect of Microwave Radiation Powers on BET Surface Area

Figure 3b shows the effect of microwave radiation powers (450, 600, 700, and 800 W) on the BET surface area of PMFC prepared at a constant activation holding time for 6 min using 2:1 activating agent ratio (PMFs char/KOH). The PMFC produced at 700 W exhibits the highest surface above 990 m^2^/g. The BET surface area is significantly influenced by microwave radiation power. Commonly, increasing surface area is harmonized with the increasing microwave radiation power. Moreover, the surface area may drop when the microwave radiation power increased continuously because more heat was produced for the activation leads to the ablation of more carbon atoms. This can cause a micropore sintering, which significantly decreased in the BET surface area [44]. From these results, the BET surface area gradually dropped when the microwave radiation power was increased to 800 W. It is possible to infer that the microwave radiation power of 700 W is an optimum condition for the production of PMFs porous carbons.

#### 3.3.3. Effect of Activation Holding Times on BET Surface Area

In Figure 3c, the relationship between activation time and the BET surface area showed that the surface area was elevated with an increment of activation time at the beginning of activation. After the activation time of 6 min, the BET surface area was gradually decreased. It could be implied that carbon atoms react with the activating agent at the active sites, indicating that the pore structure was formed when the activation time was raised from 4 min to 6 min. The BET surface areas of PMFC were obtained as 911 and 991 m^2^/g, respectively. This phenomenon also reaches the complete activation stage, leading to the maximum value of BET surface area. However, the activation still continued after operating at the activation time of 6 min. The carbon atoms were being consumed after the fully activation stage, and this revealed the development of microporous to mesoporous structure, suggesting that the BET surface area was reduced [42,44].

### 3.4. Nitrogen Adsorption/Desorption Isotherm and Pore Size Distribution

Figure 4 shows the N_2_ adsorption/desorption isotherms of PMFCs at 196 °C. All isotherm curves showed a combination characteristic of Type I(b) and Type IV(a) of adsorption isotherms, classified by the International Union of Pure and Applied Chemistry (IUPAC) [45]. The character of Type I(b) isotherm appeared to show a sharp increase of adsorbed nitrogen at a rather low relative pressure, representing that the micropores existed in the PMFC structure. Meanwhile, the Type IV(a) isotherm revealed the formation of multilayer adsorption occurring at the relative pressure between 0.2 and 1 with a hysteresis loop at a high relative pressure, caused by a capillary condensation. It can be indicated that this phenomenon was absolutely governed by mesopores in PMFCs [45]. Moreover, the pore size distribution supported the PMFCs haing a characteristic of Type I and Type IV of adsorption isotherms [26,46]. Moreover, the pore size distribution curves of PMFC samples were evaluated using the BJH model. Typically, the pore size distribution was used to determine the proportion of total pore volume, which is the relevant property of adsorbent, suggesting that the adsorbents were appropriate for the unique utilization. The pore was separated into three categories, in which the micropore sizes are less than 2 nm, mesopore sizes are 2 to 50 nm, and macropore sizes are greater than 50 nm [47]. In Figure 4, the observed pore sized distribution was very narrow. The main distribution range was within 6 nm including the micropores and mesopores observed in PMFCs. Micropore size distribution ranged between 1.8 and 2 nm, while mesopore size distribution was obtained with size of approximately 2.1 to 3.9 nm, respectively. It was found that raising microwave power improved the pore size, because micropores had been developed into mesopores, as seen in Figure 4a. However, an increase in activation time insignificantly shows the effect on the development of pore size distribution. These results can demonstrate that a certain amount of micropores and tiny mesopores are represented in the carbon matrix owing to a characteristic of lignocellulosic substances when conversed to a porous carbon. On the other hand, the evidence of KOH as an activating agent is also selective to highly developed micropores [42,48].

Nonetheless, Table 2 showed the textural pore characteristics of PMFCs produced under identical conditions. The total pore volume and mesopore volume are significantly enhanced by an increment of microwave radiation power. The total pore volume increased from 0.37 to 0.49 cm^3^/g when the microwave radiation power rose from 450 W to 700 W. Meanwhile, the total pore volume decreased after 700 W (0.46 ± 0.024 cm^3^/g). This is because the micropores are formed by the decomposition of volatile substances at the beginning. Moreover, an increase in mesopore volume was found at a higher microwave radiation power, which exhibits the pore development from micropores to mesopores, resulting in pore widening [49]. However, an increment of mesopores gradually decreased the BET surface area [48,50]. Cheng et al. [51] also revealed the same trend between an increase of microwave power and pore volume of activated carbon produced from mushroom roots using phytic acid activation. In this sense, when compared with the activation time, the results demonstrated that a longer holding time in activation also reduced the total pore volume and BET surface area. As seen in Table 2, all PMFC samples are microporous combined mesoporous. Moreover, the micropore volume was higher than the mesopore volume (>55%). This can indicate that the produced PMFCs have a microporous characteristic.

### 3.5. Proximate and Ultimate Analysis of PMFCs

According to the ultimate analysis, a high proportion of oxygen (52.61%) is represented owing to moisture content and volatile compositions in PMFs. Meanwhile, Table 3 displays the proximate and ultimate analysis of carbonized PMFs and PMFCs. The results were represented on an as-received basis. The ultimate analysis illustrated an elemental composition including C, H, N, and O. The compositions of carbonized PMFs were 68.71% C, 1.27% H, 0.95% N, and 29.07% O, respectively. The carbon percentage gradually increased from 43.73% to 68.71% by a carbonization of raw PMFs because of the devolatilization and decomposition of oxygenated compounds in the lignocellulosic biomass [52]. However, carbonization of raw feedstock is essential dramatical. That is because of the interaction for heat generation, which was provided from the reaction between a dielectric material (carbon material) and microwave. Thus, conversion of raw biomass into a char was required for microwave assisted activation process [13]. As seen in Table 3, N element was found with a small proportion from biopolymer, such as enzymes, as well as the effect of nitrogen fertilizer left in PMFs structure. Nevertheless, it may be influenced by the chemisorption during carbonization [20,33]. It was found that PMFCs have N below 1.1%. Meanwhile, the small amount of H percentage (<1.18%) is affected by moisture content, which may adsorb in sample structure, and chemical bonding with carbon atoms. Furthermore, H content existed by a hydrogen group in KOH structure from activation [15]. The PMFC produced at microwave power of 450 W for 6 min with an activating agent ratio of 1:1 has the highest H content (1.18%), and the lowest H content (0.79%) was also observed from the PMFC produced at microwave power of 700 W for 10 min with an activating agent ratio of 2:1. The C content of PMFC samples ranged from 75.27% to 80.13%. A low activating agent ratio exhibited a high C content. In general, a rise in either microwave power or activation time shows an increase in C because of a complete elimination of oxygenated compounds [16]. In addition, C percentage was elevated from 76.59% to 80.13% with microwave radiation power increasing from 450 W to 800 W. On the other hand, increment of activation time from 4 min to 6 min also increased the C percentage from 78.59% to 79.93%.

Moreover, the proximate analysis is represented in Table 3. It can be separated into moisture, volatile matter, fixed carbon, and ash, respectively. The fixed carbon and ash content are the main parameters in the proximate analysis because, with carbon atom and inorganic molecules, ash plays a significant role in various utilizations [48]. The same trend in a recent study showed that the fixed carbon and ash gradually increased with an elevated microwave radiation power. When the activation time increased, the fixed carbon and ash content also increased owing to the ablation of volatile matter. Typically, porous carbons contained the fixed carbon over 70% and low ash content between 2% and 10%. However, the high presence of ash significantly reduced the porosity, leading to the decrease in the specific surface areas from the carbon matrix [32,34]. However, a high ash content might affect the catalysis pathway, which can cause some prominences as a promoter of catalyst [43,53]. It can be found that the fixed carbon and ash content increased with the increase in microwave radiation power (450 W to 800 W) from 73.47% to 76.62% and 7.62% to 9.05%, respectively. The highest fixed carbon (76.62%) was observed from PMFC obtained at microwave radiation power of 800 W using an activating agent ratio of 2:1. Meanwhile, the second highest fixed carbon (74.56%) provided the highest surface area of 991 m^2^/g.

### 3.6. Fourier Transform Infrared Spectroscopy (FTIR) and Raman Spectroscopy Analyses

The surface chemistry of PMFCs were evaluated using the FTIR technique, which employed the explanation of different samples of raw PMFs, carbonized PMFs, and PMFCs compared with a commercial activated carbon supplied from Biosis Co., Ltd. FTIR spectra of PMFCs, recorded from the intensity between 400 and 4000 cm^−1^, were measured by direct transmittance, as displayed in Figure 5. The band centered at 3680–3000 cm^−1^ corresponds to –OH stretching, indicating the characteristic of the hydroxyl and carbonyl groups form lignin structures in lignocellulosic material. Meanwhile, the vibrations of bands at 2925 and 2850 cm^−1^ are assigned to aliphatic C–H stretching, which identify the composition of polysaccharide in cellulose and hemicellulose, respectively. Moreover, the band centered between 1550 and 1730 cm^-1^ is ascribed to carbonyl groups (C=O) of hemicellulose and lignin [12,53]. Meanwhile, aromatic –C=C– stretching of lignin was observed at the peaks of 1600 and 1512 cm^−1^. Moreover, lignin regions were found at 1380–1240 cm^−1^ and 1460 cm^−1^. The peaks of –C–O– vibrations, mentioned for a cellulose and hemicellulose, were observed at the spectra between 1200 and 950 cm^−1^. On the other hand, the intense band below 500 cm^−1^ was necessarily described to the vibration structure of inorganic molecules [53]. However, the relative intensity of FTIR spectra of samples was significantly different as a result of several experimental conditions. It was found that the band intensity of carbonized PMFs decreased from the observed FTIR spectra of raw PMFs. That is because the surface functional groups were decomposed by the degradation of lignocellulosic components at a high temperature during microwave heating [12,53]. When compared with commercial activated carbon, the FTIR spectra of PMFC are insignificantly different. The FTIR spectra shows the significant effectiveness representing a low ash content, which does not determine the relative intensity below 500 cm^−1^ in the FTIR spectra of PMFC. Furthermore, microwave-assisted KOH activation could reduce the oxygenated compounds of raw feedstocks for porous carbon production. Figure 5b showed the Raman spectra of carbonized PMFs and PMFC700-6-2:1. The D-band intensity refers to the defects and disordered carbon structures, while the G-band is generated from the graphitic carbon structure. The relative intensity of the D-band and G-band, which is proportional to the ratio of defect sites in the carbon structure, was observed at approximately 1350 cm^−1^ and 1600 cm^−1^, respectively. The values of I_D_/I_G_ of the carbonized PMFs and PMFC700-6-2:1 (the highest S_BET_) were 1.13 and 1.02, respectively, indicating that PMFC700-6-2:1 exhibits a slightly higher contribution of graphitic structures.

### 3.7. X-ray Photoelectron Spectroscopy (XPS) Analyses

The surface chemistry composition of carbonized PMFs and PMFC with the highest specific surface areas (991 m^2^/g) was accomplished using XPS analysis. The obtained XPS spectra of prior to and after activation samples can be fitted in two enriched component peaks of carbon and oxygen corresponding to C1s and O1s spectra, as can be seen in Figure 6. Accordingly, the C1s spectra of the carbonized sample and PMFC are displayed in Figure 6a,c. The high resolution of C1s peak of carbonized PMFs can be revolved into three component peaks representing the peak of graphitic carbon (–C=C–, 284.75 eV), the groups of carbon in alcohol and/or ether linkage (C–O–C, 286.23 eV), and carbon in carbonyl group (O–C=O, 288.79 eV), with relative percentages of 75.76%, 19.50%, and 4.74%, respectively. Moreover, the C1s of PMFC exhibits 89.25% graphitic carbon (–C=C–), 2.43% C–O–C groups, and 8.32% O–C=O groups, respectively. The groups of C–O–C in carbonized PMFs are still a high fraction owing to incomplete devolatilization of oxygenated compounds at the mind carbonization temperature (500 °C). After the activation, the proportion of C–O–C tends to decrease, suggesting the elimination reaction of oxygenated components could occur and evaporate from the sample surface at a high temperature during the microwave-assisted activation stage; hence, pure carbon is presented [54]. These results are assured by proximate analyses, as seen in Table 3. On the other hand, O1s spectrum exhibits the two relevant spectra representing C–O in phenols and ethers groups and the organic C=O in carboxylic acid and/or ester group centered at 531.79 and 533.63 eV, respectively, as displayed in Figure 6b,d. Meanwhile, the O1s of PMFC was fitted into three components, that is, C–O in phenols and ethers groups of 73.08% (531.38 eV), the organic C=O in carboxylic acid and/or ester group of 21.80% (532.87 eV), and the high presence of ash of 5.11% in metal oxides (529.91 eV). The results from XPS analysis could possibly be used to indicate the significant different proportions of carbon and oxygen with agreement correlation of FTIR, proximate, and ultimate analyses.

Table 4 displays the amount of nanoporous carbons (NPCs) produced from different manufacturing processes and different conditions. Various agricultural wastes, such as cherry stone, bamboo, date seed, palm wood, tomato solid waste, and empty fruit brunch, were successfully used for production of NPCs in conventional reactors (i.e., kiln and tube furnace). However, these processes commonly take a long resident time in the production. In addition, the utilization of oil palm male flowers was developed to NPCs by hydrothermal treatment (HTC) [31]. Hydrothermal was also conducted with a long processing period. Moreover, the NPCs from HTCs have rather low porosity. Therefore, it has not been widely applied for NPCs. Recently, some researchers investigated microwave heating for the conversion of agricultural wastes to NPCs. They show some advantages, such as the reduction of activation time and high porosity NPCs. The present study shows the use of oil palm male flowers, which is highly cost effective and an abundant feedstock in NPCs’ production using microwave-assisted activation. Moreover, we show that oil palm male flowers derived NPCs have a high porosity. The NPCs produced in this study can be considered a functional material for promising applications, such as adsorbent, catalyst support material, electrode for supercapacitor, and fuel cell. 

## 4. Conclusions

In this study, the results revealed that oil palm male flowers (PMFs), an abundant agricultural waste, were successfully employed as an alternative precursor for nanoporous carbons production as they provide some advantages such as being highly cost effective and eco-friendly. The desired optimum condition to produce PMFCs is considered as the microwave radiation power of 700 W with 6 min activation holding time using an activating agent ratio of 1:2. The highest surface area is exhibited over 991 m^2^/g with 0.49 cm^3^/g total pore volume at an optimum condition. On the other hand, the PMFC performed at a maintained condition of 700 W microwave radiation power with 4 min activation holding time is preferable. That is because the second highest BET surface area is approximately 911 m^2^/g, which shows the activation time reduction. These results would make it possible to utilize the as-prepared PMFC for further applications such as a functional material in catalysis pathways and even low-cost adsorbent for contaminant elimination.

## Figures and Tables

**Figure 1 materials-13-02876-f001:**
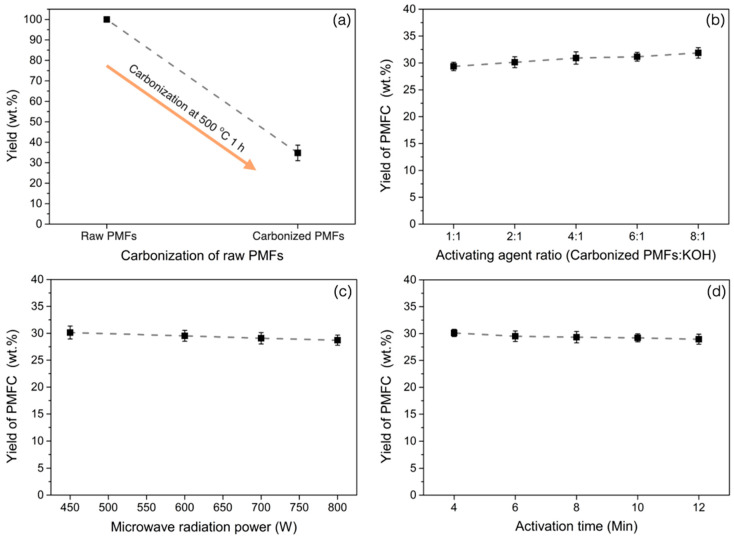
Effects of preparation variables on palm male flowers (PMFs) derived nanoporous carbon (PMFC) yield: (**a**) yield of carbonized PMFs at 500 °C, (**b**) effect of activating agent ratios, (**c**) effect of microwave radiation powers, and (**d**) effect of activation times. KOH, potassium hydroxide.

**Figure 2 materials-13-02876-f002:**
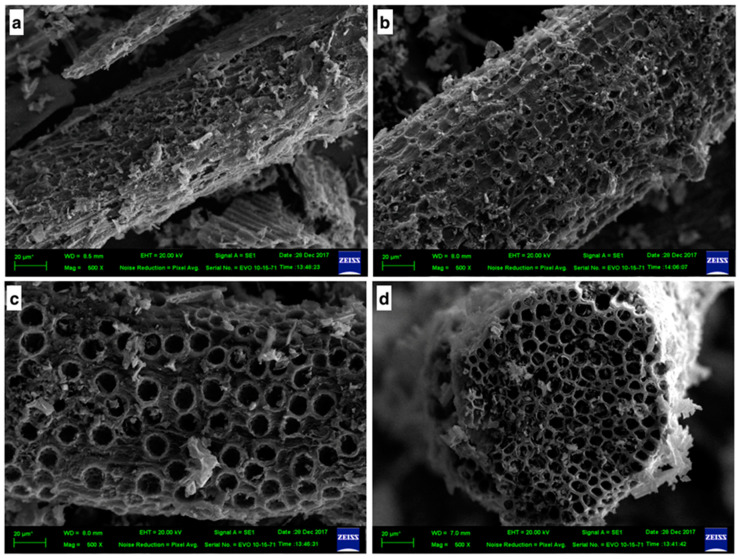
Scanning electron microscope (SEM) micrographs 500 μm: (**a**) PMFs char, (**b**) PMFC obtained at microwave power of 700 W for 4 min using 2:1 activating agent ratio (S_BET_ = 911 m^2^/g), (**c**) external surface of PMFC obtained at microwave power of 700 W for 6 min using 2:1 activating agent ratio (S_BET_ = 991 m^2^/g), and (**d**) cross section of PMFCs obtained at microwave power of 700 W for 6 min using 2:1 activating agent ratio (S_BET_ = 991 m^2^/g).

**Figure 3 materials-13-02876-f003:**
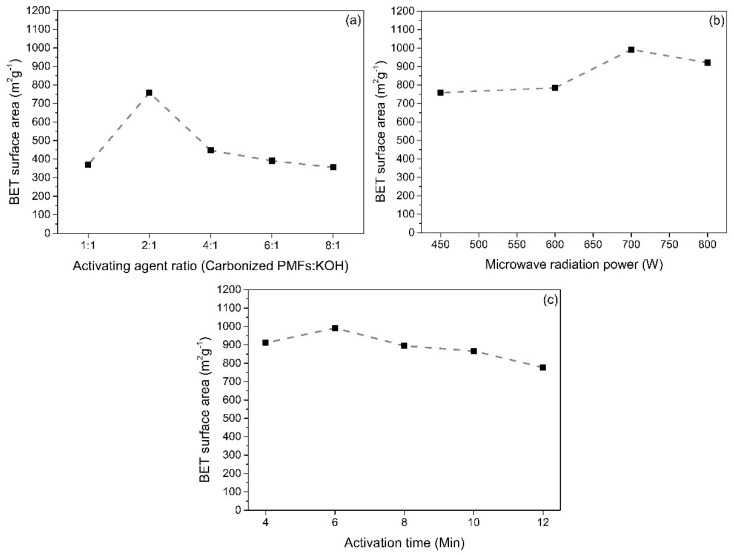
Effect of (**a**) activating agent ratio, (**b**) microwave radiation power, and (**c**) activation time on Brunauer–Emmett–Teller (BET) surface areas of PMFCs.

**Figure 4 materials-13-02876-f004:**
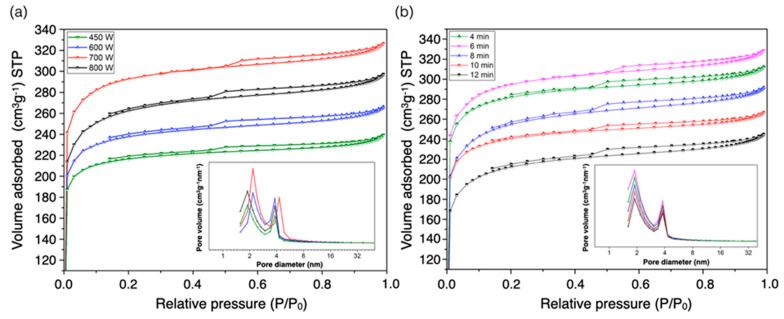
N_2_ adsorption and desorption isotherms and pore size distribution curves of PMFCs obtained at varied (**a**) microwave radiation powers and (**b**) activation holding times.

**Figure 5 materials-13-02876-f005:**
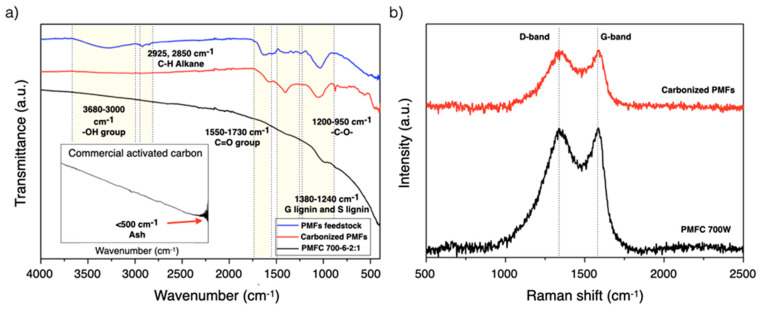
(**a**) Fourier transform infrared spectroscopy (FTIR) spectra of raw PMFs, carbonized PMFs, and PMFC and (**b**) Raman spectrum of carbonized PMFs and PMFC700-6-2:1.

**Figure 6 materials-13-02876-f006:**
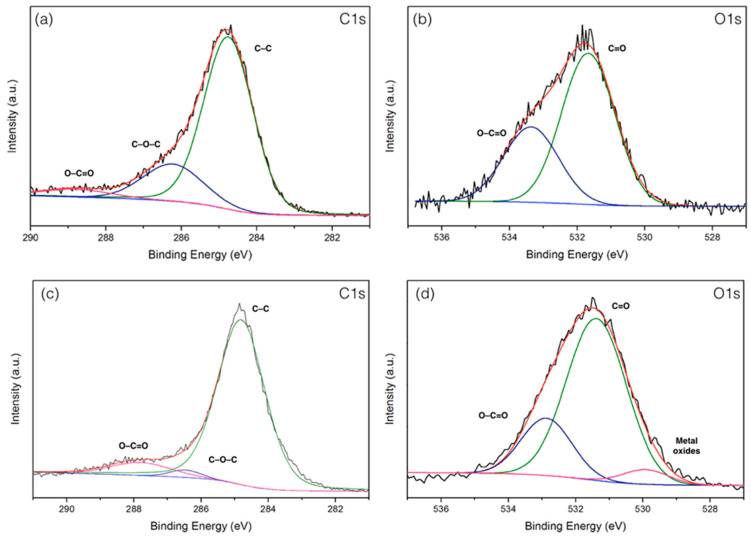
X-ray photoelectron spectroscopy (XPS) spectra of (**a,b**) carbonized PMFs and (**c,d**) PMFC-700-6-2:1.

**Table 1 materials-13-02876-t001:** Proximate and ultimate analysis of oil palm male flowers (PMFs).

Properties	Proximate Analysis **				Ultimate Analysis **			
	M	VM	FC *	A	C	H	N	O *
Raw PMFs	7.61	61.06	24.01	7.32	43.73	2.42	1.24	52.61

* Calculated by different, M: moisture, VM: volatile matter, FC: fixed carbon, A: ash, ** (as-received basis, w/w).

**Table 2 materials-13-02876-t002:** Textural pore characteristics of as-prepared PMFs nanoporous carbons. BET, Brunauer–Emmett–Teller.

Conditions	Pore Characteristics			
	S_BET_ (m^2^/g)	V_T_ (cm^3^/g)	V_mic_ (%)	V_mes_ (%)
PMFC450-6-1:1	369	0.21	57.14	42.86
PMFC450-6-2:1	757	0.37	72.97	27.03
PMFC450-6-4:1	447	0.22	77.27	22.73
PMFC450-6-6:1	390	0.19	73.68	26.32
PMFC450-6-8:1	355	0.18	61.11	38.89
PMFC600-6-2:1	784	0.39	69.23	30.77
PMFC700-6-2:1	991	0.49	71.12	28.88
PMFC800-6-2:1	920	0.46	63.04	36.96
PMFC700-4-2:1	911	0.45	73.81	26.19
PMFC700-8-2:1	895	0.44	69.38	30.62
PMFC700-10-2:1	866	0.42	64.10	36.90
PMFC700-12-2:1	777	0.39	63.64	36.36

**Table 3 materials-13-02876-t003:** Proximate and ultimate analyses of carbonized PMFs and PMFCs.

Conditions	Proximate Analysis **				Ultimate Analysis **			
	M	VM	FC *	A	C	H	N	O *
Carbonized PMFs	2.13	26.86	64.37	6.64	68.71	1.27	0.95	29.07
PMFC450-6-1:1	2.94	15.96	72.63	8.47	75.27	1.18	1.02	22.53
PMFC450-6-2:1	3.12	15.79	73.47	7.62	76.59	0.95	1.08	21.38
PMFC450-6-4:1	3.79	16.22	72.81	7.18	77.17	0.94	0.97	20.29
PMFC450-6-6:1	3.57	16.03	73.34	7.06	78.66	0.91	0.96	19.47
PMFC450-6-8:1	2.91	16.18	73.79	7.12	79.12	0.92	0.99	18.97
PMFC600-6-2:1	3.16	15.19	73.14	8.51	77.63	0.89	0.97	20.51
PMFC700-6-2:1	2.96	13.61	74.56	8.87	79.09	0.93	0.95	19.03
PMFC800-6-2:1	2.88	11.45	76.62	9.05	80.13	0.84	0.94	18.09
PMFC700-4-2:1	3.67	14.32	73.56	8.45	78.59	0.81	0.86	19.74
PMFC700-8-2:1	3.29	12.07	75.83	8.81	79.15	0.95	0.79	19.11
PMFC700-10-2:1	2.85	12.49	75.53	9.13	79.27	0.79	0.93	19.01
PMFC700-12-2:1	3.78	13.16	73.54	9.52	79.93	0.82	0.97	18.28

* Calculated by different, M: moisture, VM: volatile matter, FC: fixed carbon, A: ash, ** (as-received basis, w/w).

**Table 4 materials-13-02876-t004:** Amount of production of nanoporous carbons from different manufacturing processes and different conditions. HTC, hydrothermal treatment.

Raw Biomass	Reactor	Condition	S_BET_ (m^2^/g)	Ref.
Cherry stones	Horizontal tube furnace	CO_2_ (0.25 L/min, and KOH (KOH/char weight ratio of 2:1), 500–800 °C, 1 h	361–1173	[5]
Bamboo waste	Vertical tube furnace	Steam, 550–850 °C, 1–2.5 h	459–1210	[6]
Date seed	Hydrothermal reactor and tube furnace	Hydrothermal at 200 °C 5 h, NaOH (1:3 HTC char/NaOH w/w), activation at 600 °C 1 h	1282	[7]
Tomato solid waste	Horizontal tube furnace	ZnCl_2_ (6:1 ZnCl_2_/TW w/w), activation at 600 °C 0.5–4 h	522–1093	[8]
Palm wood	Pilot kiln	519–806 °C, 1–3.5 h CO_2_ and steam from limestone and liquefied petroleum gas (LPG) combustion	194–1084	[21]
Oil palm male flowers	Hydrothermal reactor	Hydrothermal at 180 °C 8 h, No activating agent	5	[31]
Langsat (*Lansium domesticum*) empty fruit bunch	Microwave-assisted pyrolyzer	Pre-carbonization at 700 °C, 1 h Activation at 600 W, 6 min 1.25 NaOH/char ratio (w/w)	839	[41]
Oil palm male flowers	Microwave-assisted pyrolyzer	Pre-carbonization at 500 °C, 1 h Activation at 450–800 W, 4–12 min 1:1, 2:1, 4:1, 6:1, 8:1 char/KOH ratio (w/w)	355–991	This study

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
