# Peer review of "Parametric Study on Microwave-Assisted Pyrolysis Combined KOH Activation of Oil Palm Male Flowers Derived Nanoporous Carbons"

_materials, 2020, doi:10.3390/ma13122876_

Round 1

Reviewer 1 Report

This paper describe the preparation of nanoporous carbon from waste oil palm male flowers using microwave heating.It is an interesting script, but it is not written properly. There are some points, which require minor revision and need to be clarified in the revised text. The points are described below.

  1. There are no description for Table 1 in the text. You should write the results for Table 1.
  2.  Labels of horizontal axis for Figure 1 (a) is diminished, so I can't understand this graph. Additionally, The range of vertical axis value for all graphs in figure 1 should be 0-40.
  3. In figure 3, the range of verticla axis for all graphs should be 0-1200.
  4. page 4, l. 136: "And" is deleted.

I recommended publication of this paper, subject to the above points being satisfactorily addressed.

Author Response

On behalf of the authoring team, I am very much thankful to the reviewers and editor for the deep and thorough review. We have revised our manuscript in the light of your useful suggestions and comments. Some grammatical errors and typos are also corrected. We hope our revision has improved the manuscript to a level of your satisfaction. The revised parts in the manuscript are highlighted in green.

Response to reviewer 1

1.Comment: No description for Table 1 in the text. You should write the results for Table 1

Response: Thank you for this suggestion. The description for Table 1 is added in the revised manuscript. Table 1 showed proximate and ultimate analysis of oil palm male flowers (PMFs). Proximate analyze showed that PMFs feedstocks have a fixed carbon of 24.01% and volatile matters of 61.06%. According to ultimate analysis, a high proportion of oxygen (52.61%) is represented due to a moisture content and volatile compositions in PMFs [page 10].

2.Comment: Labels of horizontal axis for Figure 1a. Additionally, the range of vertical axis value for all graphs in figure 1 should be 0-40

Response: Thank you for this suggestion. Figure 1a shows the carbonization of raw PMFs into PMFs char at 500 oC for 1 h (5 times; enough for further activation study). Figure 1a has been modified and added a label of horizontal axis. Moreover, the range of vertical axis value for all graphs in figure 1 have been changed between 0-40 [page 5].

3.Comment: In figure 3, the range of vertical axis for all graphs should be 0-1200.

Response: Thank you for this encouraging comment. The range of vertical axis for all graphs has been changed to 0-1200 [page 8].

4.Comment: Page 4, line 136: “And” is deleted.

Response: Thank you for this encouraging comment. “And” was deleted in a revised manuscript [page 4].

Reviewer 2 Report

Kaewtrakulchai et al. present their study on the MW assisted synthesis of nanoporous carbon materials using oil palm male flowers where they utilized KOH as the activating agent during the pyrolysis. Author investigated the effect of different parameters on the properties of synthesized materials aiming to provide an alternative and optimum condition for these materials. However, there are several major issues, including extensive English grammar and style improvement, in the presented manuscript which are required to be addressed and revised before I can accept the manuscript for publication in Materials.

Section 1:

There are other similar studies in the field, I would recommend to address these studies in their manuscript and describe how their work differs from them.

http://www.pjoes.com/Investigation-of-Hydrochar-Derived-from-Male-nOil-Palm-Flower-Characteristics-and,103355,0,2.html

https://www.sciencedirect.com/science/article/pii/S0143720806002245?via%3Dihub

Section 2.3.2:

Relevant references to BET, BJH, and t-plot method should be included.

Section 3.1:

What are the error margins for the calculation of yields? Figure 1 shows very slight changes (1-2 wt%) which is pronounced by limiting the y-axis range. Therefore, I think one can consider these changes in yield within the error margins, specially as they are measured on a single-sample basis.

Also, I could not realize the the x-axis of Figure 1a. It needs more clarification.

Section 3.2:

Line 186: What does tight refer to?

In Figure 2 and corresponding discussion, can authors related their discussion about the pore diameters to the diameter of pores observed in SEM images?

Section 3.4:

It is wrong to refer to the isotherms as a mix of type I and type IV. The provided isotherms in Figure 4 are obviously type IV isotherms with the hysteresis loops of type H4. Authors need to revise this session accordingly and I also recommend to add discussion from the hysteresis type.

Lines 278-279: What are the error margins for pore volume calculations? The decrease after 700 W is only 0.03 cm3/g.

Section 3.6:

In the case of presence of carbonyl groups, shouldn’t we observe strong peak around 1600-1700 cm-1?

Author Response

On behalf of the authoring team, I am very much thankful to the reviewers and editor for the deep and thorough review. We have revised our manuscript in the light of your useful suggestions and comments. Some grammatical errors and typos are also corrected. We hope our revision has improved the manuscript to a level of your satisfaction. The revised parts in the manuscript are highlighted in green.

Response to reviewer 2

1.Comment: There are other similar studies in the field, I would recommend to address these studies in their manuscript and describe how their work differs from them

“Investigation of Hydrochar Derived from Male Oil Palm Flower: Characteristics and Application for Dye Removal”

“Preparation and characterization of activated carbon from oil palm wood and its evaluation on Methylene blue adsorption”

Response: Thank you for this encouraging comment. The two publication was added in a revised manuscript in the section 1 (Introduction: Palm wood [21], Although, recent study reported that oil palm male flowers have been developed into a hydrochar via hydrothermal process. This hydrochar has a low specific surface area and poor carbon content than NPC, which may not suitable for promising applications such as an efficient adsorbent and catalyst support [31]) [page 2],

Moreover, it was compared properties with the PMFCs product, as illustrated in Table 4. “In addition, the utilization of oil palm male flowers was developed to NPCs by hydrothermal treatment (HTC) [31]. Hydrothermal also conducted with a long processing period. Moreover, the NPCs from HTCs has quite low porosity. Therefore, it has not been widely applied for NPCs” [page 14-15].

2.Comment: Material and method: Section 2.3.2: Relevant reference to BET, BJH, and t-plot method should be included.

Response: Thank you for this suggestion. Relevant references to BET, BJH, and t-plot method were included in a revised manuscript [page 4].

3.Comment: Results and discussion: Section 3.1: What is the error margin in Figure 1 in Y-axis, and Labels of horizontal axis for Figure 1a.

Response: Thank you for this encouraging comment. The error margin in Figure 1 was added in a revised figure. Also, Figure 1a shows the carbonization of raw PMFs into PMFs char at 500 oC for 1 h (5 times; enough for further activation study). Moreover, Figure 1a has been modified and added a label of horizontal axis [page 5].

4.Comment: Results and discussion: Section 3.2: Line 186: What does tight refer to? In figure 2 and corresponding discussion, can authors relate their discussion about the pore diameters to the diameters of pores observed in SEM images?

Response: Thank you for this encouraging comment. The word “tight” was changed to “dense”, which refers to dense external surface morphology of PMFC [page 5].

The discussion about the pore diameters to the diameters of pores observed in SEM images was added in a revised manuscript: “Although, the pore cavities observed by SEM (Figure 2b-c) are a microstructure of external pores of PMFC. The PMFC has various pores and cavities on an external PMFC surface, which caused by microwave-assisted KOH activation. However, this pores structure is possibly used to identify the porous carbon structures. These extensive external pores are the entry way into internal mesopores an micropores [32]. In particular, KOH reacts the carbon atoms of PMFs char during activation leads to a formation of porous structures including micropores, mesopores and external pores. Pore structures are significantly impacted by the different operating conditions during microwave-assisted KOH activation, such as activation time, and microwave radiation power [17]. However, Figure 2d shows a highly pore cavities observed by cross section image of PMFCs. The observation on the PMFC surface characteristic is related to the textural pore characteristic of produced PMFC. Finding results revealed that high porosity represented various pores on PMFCs surface” [page 6].

5.Comment: Results and discussion: Section 3.4: It wrong refer to the isotherms as a mix of type I and Type IV. The provided isotherm in Figure 4 are obviously Type IV isotherms with the hysteresis loops of Type H4. Authors need to revise this section accordingly and I also recommend to add discussion from the hysteresis type.

Line 278-279: What are the error margins for pore volume calculations? The decrease after 700 W is only 0.03 cm3/g.

Response: Thank you for this important comment. We already checked some literatures and agreed with your review. Therefore, we revised a discussion for isotherm in a revised manuscript. “All isotherm curves have a characteristic of Type IV isotherms with a wide hysteresis loop of Type H4 isotherm at high relative pressure classified by the International Union of Pure and Applied Chemistry (IUPAC). However, a rapid increase in adsorption volume at low relative pressure, which is described by the monolayer adsorption, representing a micropore adsorption in PMFCs [44, 45]” [page 8].

Line 286: Moreover, the error margins for pore volume calculations were provided in a revised manuscript “(0.46±0.024 cm3/g)” [page 9].

6.Comment: Results and discussion: Section 3.6: In case of presence of carbonyl groups, shouldn’t we observe strong peak around 1600-1700 cm-1?

Response: Thank you for this encouraging comment. We added the bands centered between 1550–1730 cm-1 is ascribed to carbonyl groups (C=O) in hemicellulose and lignin [12, 53] in a revised manuscript [page 11-12].

Reviewer 3 Report

Authors investigated the transformation of a biomass (PMF) to a series of carbons (PMFC) with the help of microwave and KOH as chemical activator. They subsequently performed a variety of textural, compositional and structural characterization to show the proof-of-concept. All materials/methods and characterization are nicely done and helpful for broad readership. On the contrary, the English writing is done very poorly and I wish authors could spend more time before submission. I can recommend this work for publication after major revision of both grammar and technical points. Please see my comments below.

As mentioned above, extensive English and proof reading by a native English speaker is necessary before re-submission.

Line 97-99: Starting with Hence. Needs to be rephrased, It is hard to understand and does not flow.  

Line 91: Please elaborate on the word “Processed”. What kind of processing. Be specific.

Line 156: Figure 1a: The X-axis values are missing. Additionally, the title should be corrected to “Carbonization Time (Min)”.

Line 167: First sentence: two enhance word has been used. The second one can be replaced by another word. The second sentence “The production yield” should be accompanied by context.

Line 171: Until instead of till. The entire sentence should be rephrased.

Line 187-190. Very poor writing and must be rephrased for a better flow. Form technical point  of view, the porosity obtained by KOH activation is way smaller to be seen by SEM or even TEM. Based on Fig 4 and Nitrogen isotherm result the pores are below 4 nm. The scale shown in SEM is 5-10 Micron. These micron size inter particle voids are different from intra-particle pores. Please cite the SEM section of the paper below for discussion.

International Journal of Hydrogen Energy 43 (40), 18549-18558

Table 2: The BET surface area must be reported without decimal place (and replaced across the manuscript). All pore volumes must be reported in 1-2 decimal place. The Dp should be deleted because it does not represent anything and will not help any discussion. Vtotl, Micro and V meso are main textural properties parameters.

Figure 4: The y-axis unit of pore volume graphs should be corrected to cm3 g-1 nm-1. Per unit size should be considered for pore volume graphs because it is a derivative.  

Author Response

On behalf of the authoring team, I am very much thankful to the reviewers and editor for the deep and thorough review. We have revised our manuscript in the light of your useful suggestions and comments. Some grammatical errors and typos are also corrected. We hope our revision has improved the manuscript to a level of your satisfaction. The revised parts in the manuscript are highlighted in green.

Response to reviewer 3

1.Comment: Line 91: Please elaborated on the word “Processed”. What kind of processing?

Response: Thank you for this encouraging comment. “crushed and sieved” was used instead of the word “Processed” in a revised manuscript [page 3].

2.Comment: Line 97-99: Starting with hence. Need to be rephased, it hard to understand and does not flow.

Response: Thank you for this suggestion. This sentence has been rephased: “Firstly, the prepared PMFs was carbonized into a char at 500 °C with a heating rate of 10 °C/min for 1 h in a horizontal stainless-steel tube reactor to enhance a carbon content due to a greater dielectric property of carbon materials since raw PMFs shows absolutely poor interaction with microwave radiation” [page 3]

3.Comment: Line 156: Figure 1a: X-axis value are missing. Additionally, the title should be corrected to “Carbonization time (Min)”.

Response: Thank you for this suggestion. Figure 1a shows the carbonization of raw PMFs into PMFs char at 500 oC for 1 h (5 times; enough for further activation study). However, Figure 1a has been modified for better understanding and accordance with context added a label of horizontal axis [page 5].

4.Comment: Line 167: First sentence: two enhance word has been used. The second sentence “The production yield” should be accompanied by context.

Response: Thank you for this encouraging comment. The word and sentence have been revised: “BET surface area might develop by enhancement of microwave power. The porous carbon yield is gradually decreased since a higher microwave radiation power serves more heat energy for the activation” [page 4].

5.Comment: Line 171: Until instead of till. The entire sentence should be rephased.

Response: Thank you for this comment. The word “Until” is instead of “Till”, and the entire sentence has been rephased: “An increment of activation time is getting a reduction of PMFCs yield until to the activation time raising to 8 min” [page 5].

6.Comment: Line 187-190: Very poor writing and must be rephased for better flow. From technical point of view, the porosity obtained by KOH activation is way smaller to be seen by SEM or even TEM. Based on Figure 4 and nitrogen isotherm result the pores are below 4 nm. The scale shown in SEM is 5-10 Micron. These micron sizes inter particle void are different from intra-particle pores. Please cite the SEM section of the paper below discussion.

Response: Thank you for this comment. The discussion about the pore diameters to the diameters of pores observed in SEM images was added in a revised manuscript: “Although, the pore cavities observed by SEM (Figure 2b-c) are a microstructure of extensive external pores of PMFCs. However, this pores structure is possibly used to identify the porous carbon structures. These extensive external pores are the entry way into internal mesopores an micropores [32]” [page 6]

7.Comment: Table 2: The BET surface area must be report without decimal place (and replaced across the manuscript). All pore volumes must be reported in 1-2 decimal place. The Dp should be deleted because it does not represent anything and will not help any discussion. Vtot,Vmicro, and Vmeso are main textural properties parameters.

Response: Thank you for this encouraging comment. The BET surface area is reported without decimal place. Also, pore volume is reported in 2 decimal place. Moreover, the Dp has been deleted from Table 2 [page 7].

8.Comment: Figure 4: The y-axis unit of pore volume graphs should be corrected to

cm3g-1nm-1 per unit size should be considered for pore volume graph because it is a derivative.

Response: Thank you for this comment. Figure 4: The y-axis unit of pore volume graphs was corrected to cm3g-1nm-1 [page 9].

Reviewer 4 Report

Author has investigated on the parametric study on microwave-assisted pyrolysis combined KOH activation on oil palm male flowers derived nanoporous carbon. The manuscript sound novel and interesting and I would like to suggest few comments to author to consider before its publication.

  1. It would be interesting to see the Raman spectroscopy of the synthesized nanoporous carbon material.
  2. Author can provide a comparison table of BET results (Method, SBET, Pore size, etc., ) with those of published literature.

Author Response

On behalf of the authoring team, I am very much thankful to the reviewers and editor for the deep and thorough review. We have revised our manuscript in the light of your useful suggestions and comments. Some grammatical errors and typos are also corrected. We hope our revision has improved the manuscript to a level of your satisfaction. The revised parts in the manuscript are highlighted in green.

Response to reviewer 4

1.Comment: It would be interesting to see the Raman spectroscopy of synthesized nanoporous carbon material.

Response: Thank you for this encouraging comment. Raman spectra of carbonized PMFs and synthesized PMFCs with the discussion were provided in a revised manuscript “Figure 5b showed the Raman spectra of carbonized PMFs and PMFC700-6-2:1. The D-band intensity is referred to the defects and disordered carbon structures, while the G-band is generated from the graphitic carbon structure. The relatively intensity of the D-band and G-band, which is a proportional to the ratio of defect sites in carbon structure, were observed at approximately 1350 cm-1 and 1600 cm-1, respectively. The values of ID/IG of the carbonized PMFs and PMFC700-6-2:1 (the highest SBET) were 1.13 and 1.02 indicating that PMFC700-6-2:1 exhibits a slightly high contribution of graphitic structures” [page 12].

2.Comment: Authors can provide a comparison table of BET result (method, conditions) with those of published literature.

Response: Thank you for this suggestion. A comparison table of BET result (method, conditions) with those of published literatures in Table 4: “Table 4 displayed the amount of nanoporous carbons (NPCs) produced from different manufacturing processes and different conditions. Various agricultural wastes, such as cherry stone, bamboo, date seed, palm wood, Tomato solid waste, and empty fruit brunch, were successfully used for production of NPCs in conventional reactors (i.e. kiln and tube furnace). However, these processes commonly take a long resident time in the production. In addition, the utilization of oil palm male flowers was developed to NPCs by hydrothermal treatment (HTC) [31]. Hydrothermal also conducted with a long processing period. Moreover, the NPCs from HTCs has quite low porosity. Therefore, it has not been widely applied for NPCs. Recently, some researchers investigated on microwave heating for conversion of agricultural wastes to NPCs. They show some advantages, such as reduction of activation time and high porosity NPCs. The present study shows the use of oil palm male flowers, which is highly cost effective and abundant feedstock in NPCs production by using microwave-assisted activation. Moreover, we show that oil palm male flowers derived NPCs has a high porosity. The NPCs produced in this study can be considered a functional material for promising applications, such as adsorbent, catalyst supporting material, electrode for supercapacitor and fuel cell” [page 13-14].

Round 2

Reviewer 2 Report

Authors revised the manuscript according to the comments provided in the previous round of review and I recommend it to be accepted for publication. 

Author Response

On behalf of the authoring team, I am very much thankful to the reviewer for the deep and thorough review. We have revised our manuscript in the light of your useful suggestions and comments. Some grammatical errors and typos are also corrected. We hope our revision has improved the manuscript to a level of your satisfaction. The revised parts in the manuscript are highlighted in green.

Reviewer 3 Report

I would like to address one of the reviewer’s comment regarding the isotherms type. The identification of isotherms by authors in the original manuscript was 100% correct and the reviewer’s comment was absolutely wrong. I am very disappointed that this reviewer did not do his/her homework before making any comments. In fact, the isotherms depicted in Figure 4 of manuscript are the classical isotherms which are usually observed in any KOH activation papers. They are consisting of both type I and type IV. The sole type I isotherms is governed by micropores (pore below 0-2nm) and characterized by a sharp increase of uptake at very low pressure and then a flat/plate for the rest of relative pressure. The pure type IV isotherm is governed by mesopore (pores 2-50 nm) and does NOT have the initial sharp uptake. The isotherm shown in this manuscript have very high initial sharp uptake and then followed by a gradual increase and hysteresis (hysteresis is caused by capillary condensation). The pore size distribution graphs are also supporting that isotherms are a combination of Type I and Type IV. Obviously, the pore size below 2 nm have not been shown because authors probably did not go to very low-pressure points such as P/P0 = 10-5 or 10-6 when they perform N2 adsorption-desorption experiment. Not showing the pore size below 2 nm does not mean there is not pore size in this range. In fact, if the authors chose to go to lower pressure points while collecting the adsorption-desorption graphs it would have yielded valuable information on pore size distribution of pores around 1-2 nm which is missing here. However, regardless of collecting low pressure points, experts in adsorption filed would easily distinguish the presence of micrpores by the initial sharp uptake. The other point is worth mentioning here is that there is a recently updated IUPAC classification for adsorption isotherms published in 2015. The new classification has Type I(a) and Type (b) for microporous materials depends on their size which I(b) consists of broader micrpores distinguished with a knee after initial sharp intake and plateau part (like this paper). Similarly, it has two subclassification of Type IV(a) and Type IV(b) for mesoporous materials in which the former comes with hysteresis. With all detailed description here, I would like the authors to make these changes before publication.

1-Please cite the reference below for updated IUPAC classification and read the Figure 2 for and explanation. Update the isotherms type in results and discussion of the manuscript to hybrid of Type I(b) / Type IV(a) using my explanation and the IUPAC reference. This correction must be done before publication. The optional work is to repeat the adsorption/desorption by adding lower pressure points (This might change all BET and pore volume numbers but will show the pore size distribution in micrpore area). The other option is to do CO2 isotherms at 273K for narrow micrpores and overlap with the existing pore size distribution. The CO2 isotherms yields information about pores 0-1 nm not larger.

Here is the IUPAC reference:

Physisorption of gases, with special reference to the evaluation of surface area and pore size distribution (IUPAC Technical Report)

Pure and Applied Chemistry 87 (9-10), 1051-1069

I am satisfied with authors’ answer to my own comments and no further revision is needed.

Author Response

On behalf of the authoring team, I am very much thankful to the reviewer for the deep and thorough review. We have revised our manuscript in the light of your useful suggestions and comments. Some grammatical errors and typos are also corrected. We hope our revision has improved the manuscript to a level of your satisfaction. The revised parts in the manuscript are highlighted in green.

Comment: Correction of the explanation for the sorption isotherms of PMFCs

Response: Thank you for this relevant comment. The correction of isotherm explanation according to your explanation with the reference for the updated IUPAC classification was done in the revised manuscript “All isotherm curves showed a combination characteristic of Type I(b) and Type IV(a) of adsorption isotherms, classified by the International Union of Pure and Applied Chemistry (IUPAC) [45]. The character of Type I(b) isotherm appeared a sharp increase of adsorbed nitrogen at rather low relative pressure, representing the micropores were existed in the PMFC structure. While, the Type IV(a) isotherm revealed the formation of multilayer adsorption occurred at the relative pressure between 0.2 and 1 with a hysteresis loop at high relative pressure, which caused by a capillary condensation. It can be indicated that this phenomenon was absolutely governed by mesopores in PMFCs [45]. Also, the pore size distribution was supported that the PMFCs have a characteristic of Type I and Type IV of adsorption isotherm” [Page 8-9].

Reference

[45] Thommes, M., et al., Physisorption of gases, with special reference to the evaluation of surface area and pore size distribution (IUPAC Technical Report). Pure and Applied Chemistry, 2015. 87(9-10): p. 1051-1069.
